# The BET Protein Inhibitor JQ1 Decreases Hypoxia and Improves the Therapeutic Benefit of Anti-PD-1 in a High-Risk Neuroblastoma Mouse Model

**DOI:** 10.3390/cells11182783

**Published:** 2022-09-06

**Authors:** Delphine Sauvage, Manon Bosseler, Elodie Viry, Georgia Kanli, Anais Oudin, Guy Berchem, Olivier Keunen, Bassam Janji

**Affiliations:** 1Tumor Immunotherapy and Microenvironment (TIME) Group, Luxembourg Institute of Health, L-1210 Luxembourg, Luxembourg; 2Translational Radiomics Group, Luxembourg Institute of Health, L-1210 Luxembourg, Luxembourg; 3NORLUX Neuro-Oncology Laboratory, Department of Cancer Research, Luxembourg Institute of Health, L-1210 Luxembourg, Luxembourg; 4Department of Hemato-Oncology, Centre Hospitalier du Luxembourg, L-1210 Luxembourg, Luxembourg

**Keywords:** neuroblastoma, immune checkpoint inhibitors, PD-1/PD-L1, immunotherapy, hypoxia, epigenetic drugs, JQ1, combinatorial therapy, tumor vasculature

## Abstract

Anti-programmed death 1 (PD-1) is a revolutionary treatment for many cancers. The response to anti-PD-1 relies on several properties of tumor and immune cells, including the expression of PD-L1 and PD-1. Despite the impressive clinical benefit achieved with anti-PD-1 in several cancers in adults, the use of this therapy for high-risk neuroblastoma remains modest. Here, we evaluated the therapeutic benefit of anti-PD-1 in combination with JQ1 in a highly relevant TH-MYCN neuroblastoma transgenic mouse model. JQ1 is a small molecule inhibitor of the extra-terminal domain (BET) family of bromodomain proteins, competitively binding to bromodomains. Using several neuroblastoma cell lines in vitro, we showed that JQ1 inhibited hypoxia-dependent induction of HIF-1α and decreased the expression of the well-known HIF-1α downstream target gene CA9. Using MRI relaxometry performed on TH-MYCN tumor-bearing mice, we showed that JQ1 decreases R2* in tumors, a parameter associated with intra-tumor hypoxia in pre-clinical settings. Decreasing hypoxia by JQ1 was associated with improved blood vessel quality and integrity, as revealed by CD31 and αSMA staining on tumor sections. By analyzing the immune landscape of TH-MYCN tumors in mice, we found that JQ1 had no major impact on infiltrating immune cells into the tumor microenvironment but significantly increased the percentage of CD8^+^ PD-1^+^, conventional CD4^+^ PD-1^+^, and Treg PD-1^+^ cells. While anti-PD-1 monotherapy did not affect TH-MYCN tumor growth, we showed that combinatorial therapy associating JQ1 significantly decreased the tumor volume and improved the therapeutic benefit of anti-PD-1. This study provided the pre-clinical proof of concept needed to establish a new combination immunotherapy approach that may create tremendous enthusiasm for treating high-risk childhood neuroblastoma.

## 1. Introduction

Neuroblastoma (NB) is the most frequent cancer type among childhood extra-cranial pediatric solid tumors and is responsible for 13% of cancer deaths in children younger than 15 years [1,2,3]. NB leads to abdominal, pelvic, thoracic, or cervical masses [1,2,4,5,6], and metastatic NB can infiltrate bones, bone marrow, lymph nodes, liver, lungs, skin, and rarely some other organs [2,4,6].

MYCN amplification is the major genetic alteration found in 20–30% of NB [7,8] and correlates with an unfavorable aggressive phenotype and treatment failure. Other genetic alterations have also been described, including (i) germline gain-of-function mutation of ALK (anaplastic lymphoma receptor tyrosine kinase) gene, reported as a predisposition factor for familial neuroblastoma [9,10]; (ii) somatic mutations of ALK, observed in 14% of high-risk neuroblastoma [11]; (iii) loss-of-function genetic alterations of ATRX (alpha-thalassemia and mental retardation, X-linked) and TERT (telomerase reverse transcriptase) genes, respectively described in 10% and 25% of patients, [12,13]; and (iv) familial mutations of PHOX2B (paired-like homeobox 2b) gene [2,5,6,14].

The International Staging System (INSS) is currently used to classify NB in six stages (Stage 1, 2A, 2B, 3, 4 and 4S) (reviewed in [15]). NB stage 4 is the most aggressive and characterized by metastasis formation. However, NB stage 4S is a localized tumor with limited metastasis. Typically, 4S neuroblastoma spontaneously regresses in young infants (<1 year of age) by a mechanism that is not fully understood. In addition to the INSS, NB patients can be stratified into low, intermediate, and high-risk disease groups depending on the age at diagnosis, stage, histology, MYCN status, and tumor cell ploidy (reviewed in [16]). MYCN amplification remains one of the major predictors of high-risk disease [8] and poor clinical outcomes [1,4,5,6]. About 60% of NB patients develop a high-risk form despite heavy and intensive treatment. The prognosis of high-risk neuroblastoma is below 50% survival at 5 years [17].

Several therapeutic options are currently used to treat high-risk NB, including a high dose of chemotherapy followed by autologous stem cell transplant, surgery, radiotherapy, and immunotherapy [18]. However, these conventional therapies are frequently associated with severe side effects. Therefore, developing a new therapy to improve prognosis and reduce side effects for high-risk neuroblastoma is urgently needed.

Accumulating evidence recognizes that targeting hypoxia represents a therapeutic strategy for high-risk NB. HIF-1α is the major transcription factor, which is stabilized in cells in response to hypoxia. Under normoxia, HIF-1α is degraded by an O_2_-dependent mechanism involving a prolyl hydroxylase (PHD)–von Hippel-Lindau (VHL) proteasome pathway [19]. In addition, HIF-1α can be degraded by O_2_/PHD/VHL-independent mechanisms through interactions with the receptor of activated protein kinase C (RACK1) following the inhibition of heat-shock protein 90 (HSP90) [20]. The high expression of hypoxia-inducible factor (HIF)-1α was correlated with poor prognosis of NB and found to be upregulated in MYCN amplified tumors and cell lines [21]. Moreover, the expression of HIF-1α downstream target gene CA9 was reported to be associated with poor survival in high-risk NB [22]. Furthermore, it has been described that RACK1 is among the top 10 genes associated with an unfavorable prognosis in NB. RACK1 depletion negatively affects the proliferation, invasion, and migration of NB cells, indicating that, similar to HIF-1α, targeting RACK1 represents a new therapeutic option for NB [23].

Bromodomain and extra-terminal domain (BET) protein inhibitor JQ1 has recently been reported to improve the survival in pre-clinical neuroblastoma mouse models, impact several cellular pathways, including MYCN expression [22], and impair hypoxic responses in triple-negative breast cancer [24].

In this study, we investigated in vitro the impact of JQ1 on the hypoxic status of several NB cell lines by assessing the expression of HIF-1α and its downstream target gene CAIX. By using the relevant TH-MYCN pre-clinical NB model, we evaluated in vivo the hypoxic status and growth of TH-MYCN tumors treated with JQ1 alone or in combination with anti-PD-1. Our results showed that JQ1 treatment impaired the expression of HIF-1α and CAIX in NB cells and improved the therapeutic benefit of anti-PD-1 in TH-MYCN.

## 2. Material and Methods

### 2.1. Cell Culture

The murine NHO2A cell line was established from homozygous TH-MYCN neuroblastoma mice. NHO2A cells were grown in RPMI 1640 medium supplemented with 10% fetal bovine serum and 1% penicillin–streptomycin in a humid atmosphere at 37 °C and 5% CO_2_. SIMA and CHP-134 cell lines are human neuroblastoma cells derived from MYCN-amplified human stage III and stage IV neuroblastoma, respectively. These cell lines were grown in the same conditions as NHO2A cells but with 20% fetal bovine serum. SIMA and CHP-134 cells were purchased from DSMZ Germany. NHO2A cells were kindly provided by Claudia Flemming (Children’s Cancer Institute Australia for Medical Research, Sydney, Australia) and Ursula Kees (Telethon Institute for Child Health Research, Perth, Australia). For all experiments, cells were seeded in 6-well plates and treated when they reached 50% confluence. NHO2A and CHP134 cells were incubated for 24 h in hypoxic conditions (0.1% pO_2_), and SIMA cells were incubated for 48 h under the same conditions. Cells were treated with a JQ1 drug at 0.5 or 1 µM (Selleckchem, Planegg, Germany). A stock solution of JQ1 (500 mM) was prepared by dissolving JQ1 in DMSO and diluted in RPMI 1640 medium for working solutions.

### 2.2. Protein Extraction for Western Blotting

Cells were washed with DPBS (Lonza, Bornem, Belgium) and lysed with 30 µL/well of RIPA Lysis Buffer (Millipore, Darmstadt, Germany). They were supplemented with protease inhibitor (Complete Protease Inhibitor Cocktail Tablets, Roche, Basel, Switzerland) and phosphatase inhibitor cocktails 2 and 3 (Sigma-Aldrich, Hoeilaart, Belgium). Proteins (40 µg) were separated on an SDS-PAGE gel and transferred onto a nitrocellulose membrane. The following primary antibodies were used for protein detection: anti-HIF-1α mAb (D2U3T, Cell signaling, Leiden, The Netherlands), anti-CAIX (Novus, Abingdom, UK), and anti-β-actin monoclonal antibody (Sigma, Overijse, Belgium). Secondary antibodies included anti-rabbit antibodies (Jackson, Ely, UK). Tumors were harvested from euthanized mice and immediately frozen in isopentane. Tumors were reduced to small pieces without previous thawing. RIPA was added, and samples were sonicated and centrifuged. The supernatant was used for Western blotting. Proteins (80 µg) were separated on an SDS-PAGE gel and transferred onto a nitrocellulose membrane. Protein detection was performed as described above.

### 2.3. RNA Extraction, Reverse Transcription, and RT-qPCR

The culture medium was removed, cells were washed with DPBS containing calcium and magnesium (Lonza, Bornem, Belgium), lysis solution was added, and total RNA extraction was performed using NucleoSpin RNA Plus (Macherey-Nagel, Hoerdt, France) according to the manufacturer’s protocol. Reverse transcription was performed on 200 ng of total RNA for NHO2A cells and 500 ng of total RNA for Sima and CHP-134 using Maxima First Strand cDNA Synthesis Kit (Thermo Scientific, St. Leon, Germany) according to the manufacturer’s protocol. qPCR was performed on 10× diluted cDNA using SYBR Master Mix according to the manufacturer’s protocol (Takyon Low ROX SYBR Master Mix, Eurogentec, Seraing, Belgium); the qPCR was done on a ViiA 7 Real-time PCR system (Applied Biosystems, Ulm, Germany) and data were calculated with the Quant Studio Real-Time PCR software. Fold changes were calculated using the 2^−ΔΔCT^ method. The following primers were used: mCAIX and hCAIX from Qiagen; mouse 18s forward (5′-GAA TCG AAC CCT GAT TCC CCG TC-3′), mouse 18s reverse (5′-CGG CGA CGA CCC ATT CGA AC-3′), human beta-actin forward (5′-GGT GGC TTT TAG GAT GGC AAG-3′), and human beta-actin reverse (5′-ACT GGA ACG GTG AAG GTG ACA G-3′).

### 2.4. In Vivo Experiments and Transgenic TH-MYCN Mouse Model

All in vivo experimental protocols were approved by the internal ethical committee of LIH and the national authority of the country under the agreement number LECR-2018-01/LUPA2019/72. We used a transgenic TH-MYCN (tyrosine hydroxylase-MYCN) mouse model, genetically engineered to overexpress MYCN under the control of a rat TH-promoter. The overexpression of MYCN occurred in neural crest cells, which resulted in tumors closely resembling human neuroblastoma in terms of tumor localization and histology, genomic aberrations, and gene expression [25].

After ordering cryopreserved sperm from hemizygous TH-MYCN mouse (129X1SvJ) from the NCI mouse repository (strain number: 01XD2), resuscitation was performed by Janvier Labs into 129S2/SvPasOrlRj females. Pups obtained from the subsequent breeding were weaned at 3 weeks of age. Genomic DNA extraction from tissue biopsy was done using the NucleoSpin Tissue kit from Macherey-Nagel according to the manufacturer’s protocol. TaqMan PCR was performed by TaqMan™ Universal Master Mix II, using UNG Kit for mouse TERT reference gene (VIC; Quencher TAMRA, Ulm, Germany) and TaqMan human MYCN Probe (FAM; Quencher NFQ, Ulm, Germany). CopyCaller Software was used to analyze the results and define the animal’s wild type, hemizygous, or homozygous status.

### 2.5. Immunohistochemistry on Tumors

Tumors were fixed in 4% paraformaldehyde for 48 h and then embedded in paraffin. Formalin-fixed, paraffin-embedded tumor sections (5 µm thick) from untreated or treated tumors were stained using H&E, anti-CD31, and anti-aSMA antibodies. HistoWiz Company (Brooklin, NY, USA) performed tumor sections and staining.

### 2.6. Magnetic Resonance Imaging (MRI) and Images Acquisition

An MRI was performed on a 3T pre-clinical horizontal bore scanner (MR Solutions, Guilford, UK), equipped with a quadrature volume coil designed for mouse body imaging. Animals were placed prone in the cradle and maintained sleep during the duration of the scans, using 2–3% isoflurane mixed with oxygen. The body temperature was maintained at 37 °C, and breathing was monitored throughout the scan sessions. The MRI protocols and parameters used are described in Table 1.

Homozygous mice underwent regular abdominal palpation. When abdominal mass appeared, the mice were randomly assigned to the JQ1 treatment group, receiving JQ1 diluted in 10% DMSO and 10% (2-Hydroxypropyl)-β-cyclodextrin (Sigma-Aldrich) or a control group receiving the vehicle by an intraperitoneal route. The dosing and treatment schedules are reported in the figures. Anatomical series (FSE T1w and FSE T2w) were used to screen the animals and calculate tumor volumes. For relaxometry assessment, only the abdominal part of the tumor, which was visible on a T2-weighted MRI, was considered, and the tumor volume was reported in mm^3^. Relaxometry series was acquired to assess tumor physiological parameters before and after treatment. Anatomical and relaxometry data were acquired for mice just before starting treatment (day 1) and repeated as described in the figures just before sacrifice.

### 2.7. Data Analysis and Statistics

Relaxometry data were analyzed in nordicICE (NordicNeuroLab, Bergen, Norway). Maps of the relaxivity parameters T2 and T2*; corresponding relaxivity rates R2 (1/T2) and R2* (1/T2*) were derived from the respective relaxometry series. R2 and R2* are surrogate hypoxia markers [26,27]. Median values were reported for the JQ1 treatment and control groups. A Student’s *t*-test was used to assess the statistical significance between the groups, calculated in GraphPad Prism 9.4.0 (San Diego, CA, USA); *p*-values  <  0.05 were considered statistically significant.

### 2.8. Fluorescence-Activated Cell Sorting (FACS) In Vivo

Immune phenotyping of tumors was assessed by Fluorescence-Activated Cell sorting (FACS, ARIA, Becton Dickinson, Dorp, Belgium). Homozygous mice were subjected to regular abdominal palpations, and when abdominal masses appeared, mice were randomly assigned to the control and JQ1 treatment groups. Tumors were collected after 3 days of treatment with the vehicle or JQ1. Only the abdominal part of the tumor was considered.

After tumor dissociation, using a 100 µm cell strainer and a syringe piston to crush the mass into a DMEM complete medium, cells were centrifuged for 10 min at 4 °C. ACK lysis buffer (Lonza, Basel, Switzerland) was used to lyse red blood cells, and a cell count was performed. After blocking FC-receptors for 5 min with CD16/CD32, appropriate antibodies were used to stain the cells. The following antibodies were used: live or dead near IR (life technology, Carlsbad, CA, USA), BuV395 rat anti-mouse CD45 (BD Horizon, Cambridge, UK), PE-CF594 hamster anti-mouse CD279 PD-1 (BD Horizon, Cambridge, UK), BV510 hamster anti-mouse CD3e (BD Horizon, Cambridge, UK), PerCP Cy5.5 anti-mouse CD4 (eBiosciences Lif Tech, Merelbeke, Belgium), BV570 anti-mouse CD8A (Biolegend, Amsterdam, The Netherlands), BV605 hamster anti-mouse CD69 (BD Horizon, Cambridge, UK), Alexa Fluor 488 anti-mouse and rat FOXP3 (eBiosciences Lif Tech, Merelbeke, Belgium), APC-R700 rat anti-mouse CD11b (BD Horizon, Cambridge, UK), PE-Cy7 anti-mouse CD11c (eBiosciences Lif Tech, Merelbeke, Belgium), BV605 anti-mouse F4/80 (Biolegend, Amsterdam, The Netherlands), BV785 anti-mouse Ly-6G (Biolegend, Amsterdam, The Netherlands), BV421 anti-mouse Ly-6C (Biolegend, Amsterdam, The Netherlands), PerCP Cy5.5 anti-mouse CD206 (Biolegend, Amsterdam, The Netherlands), Alexa Fluor 488 anti-mouse and rat FOXP3 (eBiosciences Lif Tech, Merelbeke, Belgium), and BV605 hamster anti-mouse CD69 (BD Horizon, Cambridge, UK). Data were collected using multicolor flow cytometry, and FlowJo software was used for data analysis.

### 2.9. Survival Curves

Tumor-bearing homozygous TH-MYCN mice were detected by abdominal palpation, and T2-weighted MR images were performed to confirm the presence of NB and determine the total tumor volume (the abdominal and thoracic parts when present). When the tumor reached a volume between 400 and 800 mm^3^, the mice were randomly assigned to one of four groups untreated, treated with anti-PD-1, treated with JQ, and treated with a combination of JQ1 and anti-PD-1. Treatment dosing and schedule are reported in the corresponding figure. Isotype (InVivoMAb rat IgG2a isotype control, anti-trinitrophenol, BE0089, BioXCell, Huissen, The Netherlands) or anti-PD-1 monoclonal antibodies were administered IP at 10 mg/kg/day every 3 days. Mice were scored every day for signs of discomfort or tumor volume of >2000 mm^3^. When such parameters were observed or reached, the animal was euthanized. Tumor volume was measured using T2-weighted MR images. An MRI was performed before starting treatment and then on days 4, 8 and 15. Depending on the tumor development, images were repeated at regular intervals until reaching the endpoint volume. Mouse survival probability was defined using GraphPad Prism, and *p*-values were calculated using the log-rank (Mantel-Cox) test.

## 3. Results

### 3.1. JQ1 Impairs Hypoxic Responses in Neuroblastoma Cells In Vitro

We assessed the impact of JQ-1 on the mRNA and protein expression of HIF-1α and CAIX in three neuroblastoma cell lines. NHO2A mouse cells were derived from neuroblastoma tumors of homozygous TH-MYCN transgenic mice. CHP-134 and SIMA cells were derived from patients with neuroblastoma tumors. NHO2A cell lines overexpressed *MYCN*, whereas CHP-134 and SIMA cell lines displayed amplified *MYCN*. Together, these cell lines recapitulated the pathogenesis and features of high-risk neuroblastoma in children. As expected, our results (Figure 1A) showed an accumulation of HIF-1α protein in NHO2A, CHIP-134, and SIMA cells under hypoxia. Such accumulation was associated with an increased expression of HIF-1α downstream target gene CAIX in the three cell lines tested.

We revealed that treatment of NHO2A, CHIP-134, and SIMA cells cultured under hypoxia with JQ-1 significantly decreased HIF-1α and CAIX protein levels in a dose-dependent manner. Moreover, the mRNA levels of CAIX were significantly decreased in hypoxic cells following treatment with JQ-1 (Figure 1B). Based on these data, we believe that JQ1 impairs hypoxia in neuroblastoma cells in vitro.

### 3.2. JQ1 Treatment Reduces Hypoxia in TH-MYCN Tumors

To evaluate the impact of JQ1 on the hypoxic status of TH-MYCN tumors, we conducted magnetic resonance imaging (MRI) on homozygous TH-MYCN mice to assess the relaxation rate (R2*) value, previously reported as a surrogate marker of hypoxia [28]. The treatment schedule with the vehicle (control) and JQ1 is summarized in Figure 2A. The R2* values were determined by MRI before and after the treatment, and the tumor volume was also determined by MRI after the treatment. Figure 2B showed that on day 3, there was a decrease in the tumor volume following two treatments with JQ1 compared with the control, although the difference was not statistically significant. On day 3, following two treatments, we found that the R2* values were significantly increased in the vehicle-treated mice but significantly decreased in the JQ1-treated mice (Figure 2C–E). The R2* ratio of after to before treatment was significantly decreased in JQ1-treated mice relative to vehicle-treated mice (Figure 2F). Our data suggested that JQ1 decreases the hypoxia status of TH-MYCN tumors. These results were supported by our data in Figure 1G showing a significant decrease in the HIF-1α downstream target CAIX in JQ1 treated tumors.

It is now well established that tumor microvascular networks in the hypoxic area display several unique pathological features that can be differentiated from healthy blood vessels. Such characteristics include a high density of leaky, tortuous, and primitive micro-vessels with poor pericytes’ coverage and basement membrane [29]. Based on our data showing that JQ-1 decreased hypoxia in TH-MYCN neuroblastoma tumors, we assessed the quality and integrity of blood vessels in tumors treated with JQ-1. To address this issue, immunohistochemistry of endothelial cell marker CD31 and pericyte marker SMA was performed on the vehicle- and JQ1-treated TH-MYCN neuroblastoma tumors. Figure 2H shows the vehicle-treated tumors displaying a high density of chaotic blood vessels poorly covered by pericytes. Remarkably, JQ1-treated tumors exhibited a lower number and well-structured blood vessels, which were better structured and well-covered by pericytes. Collectively, our data provided evidence that JQ1 reduced hypoxia in neuroblastoma tumors associated with blood vessel normalization.

### 3.3. JQ1-Treated Tumors Displayed a Higher Infiltration Level of CD8^+^ PD-1^+^, Conventional CD4^+^ PD-1^+^, and Treg PD-1^+^ Cells Compared to the Control

It is now well established that hypoxia impacts the tumor immune landscape [30,31,32]. Blood vessel abnormalities associated with hypoxia limit or prevent the extravasation of cytotoxic immune cells [33]. Based on our data showing that JQ1 decreased the hypoxia status of TH-MYCN tumors and induced blood vessel normalization, we evaluated the immune infiltration in the control and JQ1-treated tumors. TH-MYCN mice bearing tumors were treated as described in Figure 3A. Tumors were collected on day 4 after three treatments with either vehicle or JQ1 and processed for FACS analysis. To avoid a potential bias in the interpretation of immune infiltration, which could be associated with intra- and inter-tumor heterogeneity and volume, we performed T2-weighted MR images at the time of euthanasia and selected tumors that had a similar average volume and mass to be included in the two groups (untreated and JQ1-treated) (Appendix A).

We assessed by FACS the infiltration of lymphoid cells (CD4, CD8, and Treg) and myeloid cells (DC and macrophages). In addition, we analyzed the expression of activation and exhaustion markers CD69 and PD-1 on lymphoid cells. Our results revealed no significant difference in the infiltration of total CD8^+^, conventional CD4^+^, and neither Treg cells (Figure 3B) nor the infiltration of DC and total macrophages (Appendix A) in the either control or JQ1-treated tumors. Moreover, CD8^+^, CD4^+^, and Treg cells infiltrating the control and JQ1-treated tumors expressed a similar level of the activation marker CD69 (Figure 3C). We showed that JQ1-treated tumors exhibited a significantly higher level of CD8^+^ PD-1^+^, CD4^+^ PD1^+^, and Treg PD-1^+^ cells compared to the control (Figure 3D).

### 3.4. Combining JQ1 Improves the Therapeutic Benefit of PD-1 in TH-MYCN Tumor-Bearing Mice

Recently, the infiltration of CD8^+^ T cells expressing high PD-1 has been established as an effective biomarker for the response to immune checkpoint inhibitor therapy across multiple cancers [34]. Consistent with this, we evaluated the impact of combining JQ1 on TH-MYCN tumor response to anti-PD-1. To address this issue, homozygous TH-MYCN tumor-bearing mice were randomly assigned to several groups for treatments with mono (JQ1 or PD-1 alone) or combination (JQ1+PD-1). Mice included in the experiments were those having developed tumor volumes ranging from 400–800 mm^3^ based on T2-weighted MR images performed before starting the treatments (Day 1). MRI images also assessed tumor volumes after treatment on days 4, 8 and 15. The different mouse groups and treatment schedules are summarized in Figure 4A. Representative images of one mouse from each group on days 1, 4, 8 and 15 are reported in Figure 4B. As shown on day 1, the tumor mass (delineated by a red line) was comparable in all groups (Figure 4B). However, the tumor volume was significantly reduced on days 8 and 15 in mice treated with a combination of JQ-1 and anti-PD-1 relative to mice treated with JQ-1 alone (Figure 4C). This reduction in the tumor volume was translated into a significant improvement in mice survival in mice treated with a combination of JQ1 and anti-PD-1 relative to those treated with JQ-1 alone (Figure 4D).

## 4. Discussion

In this study, we revealed that combining JQ1 improves the benefit of anti-PD-1 in the TH-MYCN NB mouse model. Such improvement could be related to the effect of JQ1 in decreasing hypoxia in TH-MYCN NB tumors. Indeed, JQ1 and derivatives are currently attracting major interest in treating hematological and solid cancers, including pediatric malignancies [35]. We strongly believe that the ability of JQ1 to decrease hypoxia in TH-MYCN NB tumors relies on the impairment of the transcriptional activity of HIF-1α. This concept was supported by: (i) our data showing that treatment with JQ1 decreased the mRNA and protein levels of CA9 in several NB cell lines, including those derived from TH-MYCN tumors; and (ii) previous studies showing that JQ1 impairs hypoxia in triple-negative breast cancer through its ability to interact with HIF-1α and inhibit its transcription activity [24,36].

Remarkably, in all hypoxic NB cell lines, we showed that the protein level of HIF-1α decreased following treatment with JQ1 in a dose-dependent manner. Although the exact mechanism responsible for the decrease of HIF-α protein levels remains unknown, it is tempting to speculate that this could be related to the impact of JQ1 in reactivating the ubiquitin proteasomal system (UPS) responsible for HIF-1α degradation. Additional experiments need to be performed to assess the reactivation of UPS in JQ1-treated cells and tumors.

Since CA9 overexpression is associated with poor survival in NB patients, we believe that JQ1, through inhibiting HIF-1α/CAIX axis, may inhibit the growth of TH-MYCN tumors. However, our results, depicted in Figure 1B, showed that, although there is a clear trend toward a decrease, the average tumor volume between vehicle- and JQ1-treated tumors was not significantly different. This could be related to the narrow therapeutic windows used in the experimental design or to the limited group size. Indeed, the graph depicted in Figure 2B suggests that JQ1 treatment appeared to be efficient in decreasing the volume in many tumors but not in all tumors, which may be caused by biological or experimental reasons. Furthermore, considering the role of RACK-1 in NB invasion and migration, it would be interesting to evaluate the impact of JQ1 on the regulation of RACK-1 in NHO2A mouse cells in vitro and in TH-MYCN tumors in vivo.

Nevertheless, the role of JQ1 in decreasing hypoxia in TH-MYCN tumors has also been supported in vivo by assessing the R2* values using MRI technology. Our results further supported that implementing the R2* assay in the clinic may be beneficial as this is a non-invasive method to indicate the hypoxic status in pediatric tumors where invasive procedures are always difficult to implement [37].

JQ1 displayed an anti-angiogenic effect by reducing the expression of the angiogenic pathway, the key angiogenic inducer VEGF-A, and the blood vessel count. In keeping with this, we showed that treatment of TH-MYCN tumor-bearing mice with JQ1 seemed to reduce the number of blood vessels while improving their quality and integrity, as revealed by CD31/αSMA staining. Therefore, our data supported the concept that JQ1 showed typical behavior of anti-angiogenic agents in NB tumors.

By assessing the infiltration of major cytotoxic immune cells, our data revealed no impact on the infiltration of CD8, CD4 and Tregs into the tumor microenvironment of TH-MYCN tumors following treatment with JQ1. However, JQ1 treatment significantly increased the expression of PD-1 on CD8^+^, CD4^+^ and Tregs. The role of PD-1 expression on CD8^+^ cells has been extensively evaluated. Initially described as an exhaustion marker, the expression of PD-1 on CD8^+^ T cells is now reported as a strong predictor of the response to ICB in NSCLC and correlated with increased overall survival [38]. The predictive value of PD-1^high^ CD8^+^ T cells was also reported across five cancer types in several clinical samples and mouse models (reviewed in [34]). Recently, the combined expression of PD-1 on circulating CD4^+^ and CD8^+^ T cells before ICB treatment has been considered to guide therapy for patients with NSCLC [39]. Our data showed a significant increase in CD4^+^ PD-1+ and CD8 PD-1^+^ in the microenvironment of JQ1-treated TH-MYCN tumors highlighting the value of combining JQ1 and anti-PD-1. Indeed, we showed that JQ1 synergizes with anti-PD-1 to elicit a remarkable anti-tumor effect compared with JQ1 or anti-PD-1 alone. We believe that such a synergistic result relied on the effect of JQ1 to activate CD8^+^ T cells through the upregulation of PD-1. Knowing that NFATc1 is reported to regulate the expression of PD-1 expression in activated T cells [40,41], it would be interesting to determine whether JQ1 regulates NFATc1 in CD8 T cells infiltrating TH-MYCN. Nevertheless, PD-1-overexpressing CD8 T cells engaged PD-L1 on tumor cells, which can subsequently be released by anti-PD-1. Although the underlying mechanism of PD-1 upregulation on CD4, CD8, and Tregs following JQ1 treatment has not yet been investigated, we speculate that such a mechanism relies on epigenetic regulation or an increase in the protein synthesis of PD-1 by JQ1.

Although much remains to be learned mechanistically, our in vivo data are supported by a previous report showing that combining JQ1 and anti-PD-L1 led to a synergistic effect in pancreatic cancer [42]. Moreover, cooperative effects between JQ1 and anti-PD-1 have been reported in Kras+/LSL-G12D; Trp53L/L (KP) mouse models of NSCLC. In this model, combining JQ1 with anti-PD-1 impaired the immunosuppressive activity of Tregs and favored the activation of T cells in the tumor microenvironment. The JQ1–anti-PD-1 combination induced robust and long-lasting anti-tumor responses associated with an improvement in the overall survival compared to each treatment alone [43].

Taken together, our study provided convincing data supporting the concept that combining BET bromodomain inhibition JQ1 with immune checkpoint blockade based on PD-1 offers a promising therapeutic approach for high-risk neuroblastoma displaying MYCN amplification. However, given that not all high-risk NB is MYCN-amplified, it would be interesting to evaluate whether our data described here can be translated to MYCN-non-amplified NB.

## Figures and Tables

**Figure 1 cells-11-02783-f001:**
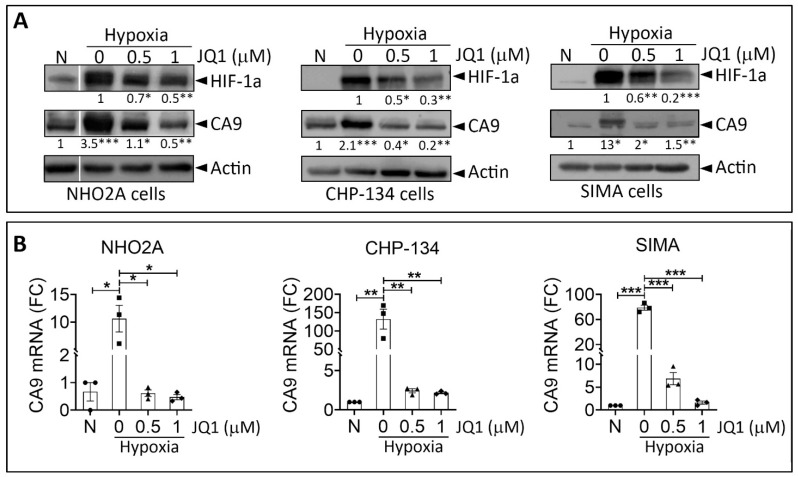
Effect of JQ1 on the hypoxia status of NB cells in vitro. (**A**) Western blot test showing the protein expression of HIF-1α and CA9 in NHO2A, CHP-134, and SIMA cells cultured under normoxia (N) or hypoxia and treated with JQ1 at the indicated concentration. Actin was used as a loading control. The quantification of band intensity corresponding to HIF-1α and CA9 in treated hypoxic cells is reported compared to untreated hypoxic cells and normoxic cells, respectively. (**B**) RT-qPCR measurement of CA9 mRNA in cells described in (**A**). Bars represent means from three independent experiments ± SEM. Statistically significant difference was calculated by unpaired two-tailed Student’s *t*-test (* *p* < 0.05, ** *p* <0.01; and *** *p* < 0.001).

**Figure 2 cells-11-02783-f002:**
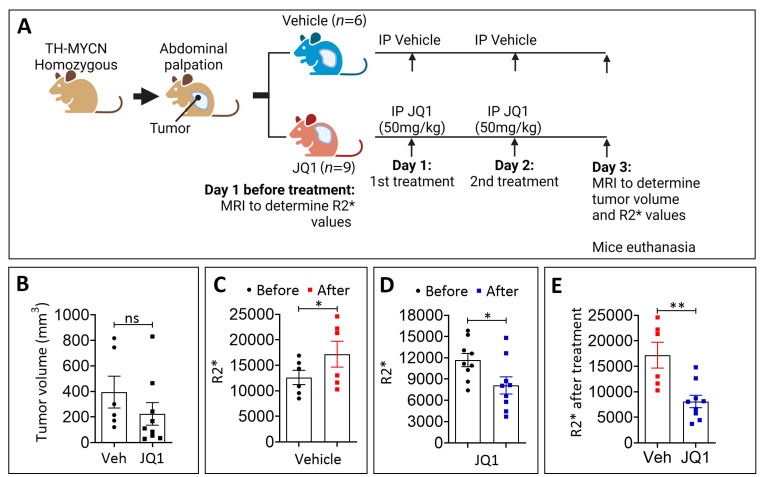
Effect of JQ1 on the hypoxia status and blood vessels in TH-MYCN tumor-bearing mice. (**A**) Experimental design of JQ1 treatment of TH-MYCN tumor-bearing mice showing the schedule and dosing. Following the development of homozygote TH-MYCN tumors, the tumor volume and R2* values were assessed by an MRI in all mice. Mice were randomly assigned to a vehicle-treated group (*n* = 6) or JQ1-treated group (*n* = 9). Mice were treated twice with either the vehicle or JQ1 (50 mg/kg) by IP on days 1 and 2. On day 3 post-treatment, an MRI was performed to determine the tumor volume and R2* values. Tumors were harvested for subsequent experiments. (**B**) The volume of TH-MYCN tumors treated with either the vehicle (Veh) or JQ1 as described in (**A**) on day 3. Each dot represents one tumor. Results are shown as mean ± SEM (error bars). Statistically significant differences were calculated compared to the Veh-treated tumors using an unpaired two-tailed Student’s *t*-test (ns: not significant). R2* values in TH-MYCN tumors before and after treatment on day 3 with the vehicle (**C**) and JQ1 (**D**) according to the experimental design in (**A**). The R2* values in TH-MYCN tumors after the treatment with either the vehicle (Veh) or JQ1 are reported in (**E**). The R2* ratio of after to before treatment with the vehicle (Veh) or JQ1 is reported in (**F**). Each dot represents one tumor. Results are shown as mean ± SEM (error bars). Statistically significant differences are calculated using a Mann–Whitney test for (**C**) and (**D**) or an unpaired two-tailed Student’s *t*-test for (**E**) and (**F**) (* *p* < 0.05, ** *p* < 0.01; and *** *p* < 0.001; ns: non-significant). (**G**) Western-blot showing the protein expression of CAIX in three different (1, 2 and 3) TH-MYCN tumors treated with either the vehicle or JQ1 according to the experimental schedule described in (**A**). Actin was used as a loading control. (**H**) Staining of vehicle- or JQ1-treated TH-MYCN tumors described in (**A**) with H&E (upper panels), CD31 (middle panels), or aSMA lower panels. Enlarged images of the zones delineated with black boxes (in CD31 and aSMA stained tumors) are shown (Scale bars 100 or 50 µm).

**Figure 3 cells-11-02783-f003:**
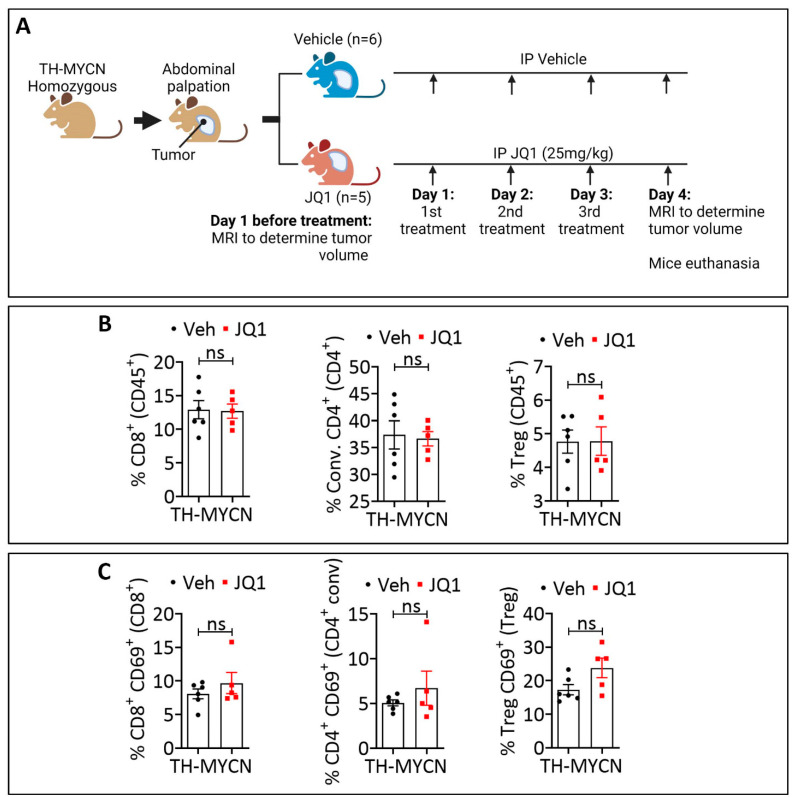
Immune phenotyping of JQ1-treated TH-MYCN tumors. (**A**) Experimental design of JQ1 treatment of TH-MYCN tumor-bearing mice showing the schedule and dosing. Following the development of homozygote TH-MYCN tumors, the tumor volume was assessed by MRI in all mice. Mice were randomly assigned to a vehicle-treated group (*n* = 6) or a JQ1-treated group (*n* = 5). Mice were treated three times with vehicle or JQ1 (25 mg/kg) by IP on days 1, 2, and 3. On day 4 post-treatment, MRI was performed to determine the tumor volume, and tumors were harvested for immune phenotyping experiments. (**B**–**D**) Flow cytometry quantification of total (**B**), CD69^+^ (**C**), or PD-1^+^ (**D**) CD8^+^ T cells, conventional (Conv.) CD4^+^ T cells and Tregs infiltrating vehicle-treated or JQ-1-treated TH-MYCN tumors at day 4. The defined subpopulations were gated and quantified in live CD45^+^ cells. Each dot represents one tumor. The data are reported as the average of six or five mice per group. Results are shown as mean ± SEM (error bars). Statistically significant differences (indicated by asterisks) are calculated compared to vehicle-treated tumors using an unpaired two-tailed Student’s *t*-test (ns: not significant *: *p* < 0.05).

**Figure 4 cells-11-02783-f004:**
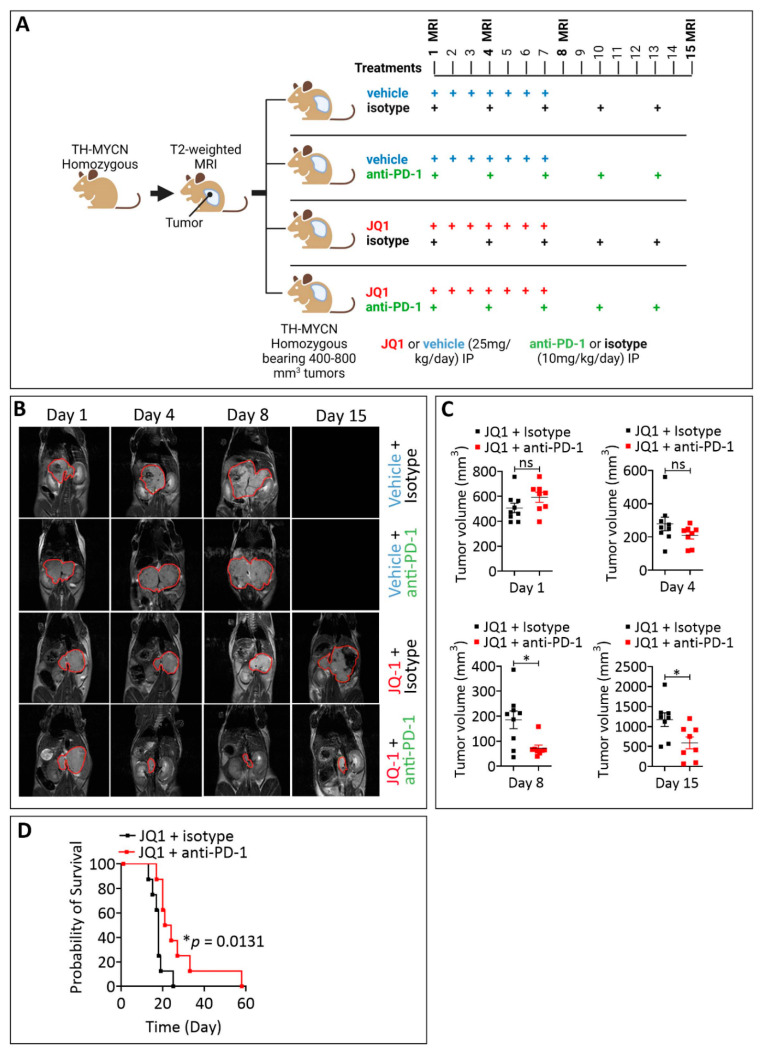
Impact of combining JQ1 and PD-1 on TH-MYCN tumor volume and mice survival. (**A**) Experimental design of JQ1 and anti-PD-1 treatment of TH-MYCN tumor-bearing mice showing the schedule and dosing. Following the development of homozygote TH-MYCN tumors, the tumor volume was assessed by MRI in all mice. Mice were randomly assigned into four groups: vehicle- and isotope-treated group (*n* = 9), vehicle and anti-PD-1-treated group (*n* = 8), JQ1 and isotype-treated group (*n* = 8), and JQ1 and anti-PD-1-treated group (*n* = 8). Mice were treated daily from days 1 to 7 with either vehicle (group 1 and 2) or JQ1 (group 3 and 4) and on days 1, 4, 7, 10 and 13 with either isotype (group 1 and 3) or anti-PD-1 (group 2 and 4). After day 15, only the treatment with isotype or anti-PD-1 was continued using the same schedule until the end of experiment (mice euthanasia). JQ1 (25 mg/kg/day) and anti-PD-1 (10 mg/kg/day) were administered by IP. MRI was performed on days 1, 4, 8 and 15 post-treatment and on a regular basis to determine the tumor volume. (**B**) Representative images of TH-MYCN tumor-bearing mice on days 1, 4, 8 and 15 for the groups described in (**A**). On day 15, no images for groups 1 and 2 are provided as animals died. Abdominal tumor masses are delineated in red. (**C**) Volumes of TH-MYCN tumors on days 1, 4 and 8, and in mice treated with JQ1 and isotype or JQ1 and anti-PD-1. Each dot represents one tumor. Results are shown as mean ± SEM (error bars). Statistically significant differences are calculated compared to the control group (JQ1 and isotype) using an unpaired two-tailed Student’s *t*-test (ns: not significant *: *p* < 0.05). (**D**) Mice survival curves were generated from tumor-bearing mice treated with JQ1 and isotype or JQ1 and anti-PD-1. Lack of survival was defined as death or tumor size > 2000 mm^3^. The probability of survival was defined using GraphPad Prism, and *p*-values were calculated using the log-rank (Mantel-Cox) test (* *p* ≤ 0.05).

**Table 1 cells-11-02783-t001:** MRI protocols used to monitor TH-MYCN tumor volumes.

Sequence	Purpose	Parameters	Number of Mice in Group Treated with	
			JQ1	Vehicle
FSE T2w	Tumor detection, delineation and volume measurement	SR: 156 µm × 161 µm × 1500 µm, Sl: 16, TE: 68 ms, TR: 5000 ms, ET: 8, AVG: 3, SD: 7 min 45 s, RG ON	9	6
FSE T1w	Tumor detection, delineation and volume measurement	SR: 156 µm × 159 µm × 1500 µm, Sl: 16, TE: 11 ms, TR: 1000 ms, ET: 4, AVG: 4, SD: 4 min 12 s, RG ON		
MEMS	T2 relaxometry map	SR: 156 µm × 208 µm × 1500 µm, Sl: 16, fTE: 15 ms, NOE: 10, TR: 3000 ms, AVG: 1, SD: 9 min 36 s, RG ON		
MGE	T2* relaxometry map	FLASH, SR: 313 µm × 313 µm × 1500 µm, Sl: 16, fTE: 4 ms, TR: 600 ms, NOE: 10, AVG: 1, SD: 1 min 16 s, RG ON		

SR: spatial resolution, Sl: number of slices, TE: echo time, TR: repetition time, ET: echo train, AVG: number of averages, SD: scan duration, RG: respiratory gating, fTE: first echo time, NOE: number of echoes, FLASH: fast low angle shot, NE: number of experiments, TRes: time resolution.

## Data Availability

Not applicable.

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
