# Peer review of "The BET Protein Inhibitor JQ1 Decreases Hypoxia and Improves the Therapeutic Benefit of Anti-PD-1 in a High-Risk Neuroblastoma Mouse Model"

_cells, 2022, doi:10.3390/cells11182783_

Round 1
Reviewer 1 Report
The manuscript deals with TH-MYCN tumours induced in mice that have been analysed by expression and function of JQ1 in increased the percentage of CD8+ PD-1+, CD4+ PD-1, Treg PD-1+. JQ1 inhibits hypoxia-dependent induction of HIF-1 and decreases the expression of the well-known HIF-1 downstream target gene CA9. The authors use both cell neuroblastoma lines and abdominal-induced tumours. The manuscript is potentially interesting, however, it needs some refinements in the introduction but also in reference lists, and discussion.
Introduction:
Lines 90-95 “In this study, we evaluated the therapeutic benefit of anti-PD-1 in combination with 90 JQ1 using the relevant TH-MYCN pre-clinical NB model. This mouse model is well-defined for pre-clinical testing of therapies aimed at treating high-risk MYCN-amplified NB. We used MRI relaxometry to reveal that JQ1 significantly decreased hypoxia in NB tumours. Furthermore, we showed that combining JQ1 improves the therapeutic benefit of anti-PD-1 in TH-MYCN”
The aim should be well defined with the two models in vivo and in vitro.
Lines 57-81 This part of the introduction is too long, the authors should summarize this part in a few lines.
However, the introduction and Discussion sections lack some important questions regarding the synthesis of some key proteins and the role of RACK1.
When suffering the hypoxic environments, cancer cells regulate a series of gene expressions and the corresponding pathways essential for cell survival and stress adaptation. Hypoxia-inducible factor 1α (HIF-1α) serving as one of the principal transcription factors is increased by hypoxia and accounts for the regulation of the expression of hypoxia-response genes. Under hypoxia, HIF-1α protein expression rapidly accumulates and regulates downstream target gene expression. The degradation of HIF-1α is also regulated by an oxygen-independent mechanism involving HIF-1α binding to the receptor of activated protein kinase C (RACK1) and Heat Shock Protein 90 (HSP90). The role of hypoxia in cancer is underexposed in the introduction and also the role of hypoxia in translation and the role of ribosomal RACK1 in regulating neuroblastoma [Mol Cell. 2007 Jan 26;25(2):207-17. doi: 10.1016/j.molcel.2007.01.001. PMID: 17244529; Cellular Signalling 53 (2019) 102–110, https://doi.org/10.1016/j.cellsig.2018.09.020
Material and methods
Lines 171- 175. Homozygous mice underwent regular abdominal palpation, and when abdominal mass appeared, mice were randomly assigned to the JQ1 treatment group, receiving JQ1 diluted in 10% DMSO and 10% (2-Hydroxypropyl)-β-cyclodextrin (Sigma-Aldrich) or a control group receiving vehicle by an intraperitoneal route. The dosing and the treatment schedules are reported in the figures. Anatomical series (FSE T1w and FSE T2w) were used to screen the animals and calculate tumour volumes... and Line 182. "Tumour volume was defined as the part of the abdominal tumour visible."
The authors should explain in the materials and methods section that the neuroblastoma study is on a neuroblastoma metastasis.
Results
The authors should consider the role of protein synthesis in the involvement of PD-1 expression (Nat Commun 8, 511 (2017). https://doi.org/10.1038/s41467-017-00612-6)
Author Response
We would like to thank the reviewer for the constructive comments. We have carefully and comprehensively addressed point-by-point all the comments and modified the text accordingly in the revised manuscript. All changes are highlighted in yellow. Please see the attached document.

Reviewer 2 Report
This paper evaluates the role of JQ1, a bromodomain inhibitor, in increasing sensitivity of MYCN amplified neuroblastoma cells to PD1 inhibition.
The paper is laid out nicely, reviewing the role of HIF1 and hypoxia in neuroblastoma and other cancers.
Were these experiments performed in any MYCN non-amplified cell lines? Given that not all high risk neuroblastoma is MYCN-amplified, it would be helpful to see a favorable response at the level of cell lines to show generalizability of the regimen before suggesting a trial in humans.
Author Response

(The authors gave the same response as above.)

Round 2
Reviewer 1 Report
The authors have followed the suggestion and now the manuscript, in my opinion, is editable.